# Information about confirmatory studies required for new drugs conditionally approved by Health Canada: A cross-sectional study

Joel Lexchin[1,2,3]*

**1** School of Health Policy and Management, York University, Toronto, Ontario, Canada, **2** Department of Family and Community Medicine, University of Toronto, Toronto, Ontario, Canada, **3** University Health Network, Toronto, Ontario, Canada

\* jlexchin@yorku.ca

**Data Availability Statement:** All relevant data are within the paper and its Supporting information files.

**Funding:** The author received no specific funding for this work.

## Abstract

### Background

Health Canada conditionally approves new drugs using its Notice of Compliance with conditions (NOC/c) policy. Under this policy Qualifying Notices (QNs) list confirmatory studies that need to be conducted to confirm the drug's efficacy. This study examines the depth of information about methodology and patient demographics in the confirmatory studies. It also compares the outcomes (surrogate or clinical) used to approve the drugs with the outcomes proposed in the confirmatory studies.

### Methods

A list of drugs approved under the NOC/c policy and their QNs were sourced from two previous publications as well as Health Canada's NOC/c website. Patient demographics and study methodology in the confirmatory studies listed in the QNs was recorded and counted. The primary outcome used to approve new drugs was recorded from Health Canada's Summary Basis of Decision website and compared to the type of outcome for studies mentioned in the QNs.

### Results

Seventy-eight drugs were approved using a NOC/c from the time the first drug was approved under the program in July 1998 until May 18, 2022. QNs were missing or all information was redacted for 3 drugs, the remaining 75 QNs listed 154 studies (median of 2 studies per QN, interquartile range 1,3). The outcome, randomization and blinding could not be determined for any study in 43 (57.3%), 36 (48.0%) and 42 (56.0%) QNs, respectively. No study gave the distribution of men and women and the number of patients was given in 23 (14.9%) studies. The expected time of completion of the studies was available for 36 (23.4%) out of 154 and information to identify studies was present for 77 (50.0%), absent for 23 (14.9%) and unclear for 26 (16.9%). Surrogate outcomes were used to approve 54

**Competing interests:** In 2019-2021, Joel Lexchin received payments for writing a brief on the role of promotion in generating prescriptions for Goodmans LLP and from the Canadian Institutes of Health Research for presenting at a workshop on conflict-of-interest in clinical practice guidelines. He is a member of the Foundation Board of Health Action International and the Board of Canadian Doctors for Medicare. He receives royalties from University of Toronto Press and James Lorimer & Co. Ltd. for books he has written. This does not alter my adherence to PLOS ONE policies on sharing data and materials.

(84.4%) of 64 drugs. Eight (14.8%) confirmatory studies for these 54 drugs used clinical outcomes, 15 (27.8%) used surrogate outcomes and outcomes were unknown for 31 (57.4%). Specifically for oncology drugs, 44 were approved with surrogate outcomes and one with a clinical outcome. Eight (18.2%) of the 44 oncology drugs approved with surrogate outcomes had confirmatory studies that used clinical outcomes, 14 (31.8%) used surrogate outcomes and the outcome could not be determined for 22 (50.0%). The sole oncology drug approved with a clinical outcome had a confirmatory study with a surrogate outcome.

## Discussion

QNs contain little information about the methodology or patient demographics of confirmatory studies. Confirmatory studies with surrogate outcomes were used almost one-third of the time to validate efficacy in drugs initially approved using surrogate outcomes. Health Canada needs to develop a template about what information regarding confirmatory studies should be contained in a QN and rethink its use of confirmatory studies using surrogate outcomes. All the data were gathered by a single individual possibly introducing unintended biases.

## Introduction

Health Canada conditionally approves new drugs (new molecules that have never been marketed in Canada before) through its Notice of Compliance with conditions (NOC/c) pathway [1], a pathway that was initiated in 1998. The goal is to "provide patients suffering from serious, life threatening or severely debilitating diseases. . .with earlier access to promising new drugs" [2]. Most approvals are based on promising surrogate outcomes that "are reasonably likely, based on available evidence, to predict an effect of a drug on recognized clinical outcomes such as morbidity and mortality" [2].

Once Health Canada has determined that the evidence submitted by a manufacturer for a new drug is satisfactory, it issues a NOC/c-Qualifying Notice (QN) which outlines the additional clinical evidence to be provided in confirmatory studies and the timelines for these studies that will verify the clinical benefit of the drug. The company marketing the drug must respond with a Letter of Undertaking (LoU) which contains details on how it will meet the conditions. Health Canada then finalizes the conditions specified in the LoU and issues a Notice of Market Authorization with Conditions and in addition makes the QN publicly available [2]. (Prior to February 2003 the QN did not exist. In this study QN is used to refer collectively to QNs and LOUs.)

Before the confirmatory studies are completed, the conditions in the NOC/c are listed as unfulfilled. Once the studies are completed, they are evaluated by Health Canada similar to studies submitted in the premarket period and if they are acceptable to Health Canada [3], the conditions are deemed to have been fulfilled and the product receives a full NOC. Should these postmarket trials not provide sufficient evidence of clinical benefit the NOC/c could be revoked and the product removed from the market and the conditions would also not be fulfilled [4].

To date, the literature examining this regulatory pathway has primarily looked at the additional therapeutic value [5] and postmarket safety [6] of drugs and how long it takes for the

conditions to be fulfilled [7], but there has not been any systematic examination of the description of the confirmatory studies in the QNs.

The Health Canada guidance document does not outline what information the QNs should contain. Therefore, this study adopts the perspective of clinicians in examining the type of information about the confirmatory studies in QNs. Specifically, will these studies provide clinicians with knowledge that will help resolve uncertainties about the benefits of drugs with a NOC/c: a) what are the demographics of the patients who will be enrolled; b) what methodology will the studies employ (e.g., randomization and blinding); and c) if the drugs were given a NOC/c based on surrogate outcomes what outcomes will the confirmatory studies use. In addition, are two other pieces of information that are potentially important to clinicians in the QNs: a) when will the studies be completed so that they can take necessary precautions in prescribing until that point; and b) is there enough identifying information about the studies that will enable them to search for and read the studies after they have been completed.

## Methods

### Source of NOC/c and QN

The generic name of all new drugs approved with a NOC/c along with the dates of granting and fulfillment of the NOC/c from the time that the first drug was approved under the program in July 1998 until May 18, 2022 were sourced from the Notice of Compliance with conditions website (http://www.hc-sc.gc.ca/dhp-mps/prodpharma/notices-avis/conditions/index-eng.php), supplemented by information from the articles by Lexchin [4] and Law [8]. QNs were available on the NOC/c website, however they are removed from the website after the postmarket commitments have been fulfilled. In that case an Access to Information (ATI) request was filed or for NOC/c issued before QNs were used the Letter of Undertaking was requested through an ATI.

### Information extraction

The following information about the confirmatory studies was extracted from the QNs:

1. Study methodology: outcome to be used (surrogate or clinical), randomization, double-blinding, length of study;

2. Patient demographics: number of men and women, age, number of patients;

3. Other: expected date of completion, study identification information.

Only the primary outcome of studies was recorded. If the date of expected completion of a study only gave the month and year, then the last day of the month was used; if the date was the given as a "quarter" of a year then the last day of the quarter was used. At times it was unclear if a trial identification number was an internal company number or a number assigned by an external body (for example clinicaltrials.gov) and in that case identification was recorded as "unclear". If the indication was withdrawn or the drug discontinued by the company, reasons for this decision were recorded from the NOC/c webpage.

All drugs were categorized at the second level of the World Health Organization (WHO) Anatomical Therapeutic Chemical (ATC) classification system [9].

QNs for new indications for existing drugs were not analyzed for a number of reasons: a) Health Canada does not provide the date when the application for the new indication was filed; b) there is no publicly available source about whether surrogate or clinical outcomes were used in granting a NOC/c for the new indication; and c) when a new indication is granted

for existing drugs clinicians have already had some experience prescribing the drug and therefore the information in the confirmatory studies may be less important for them.

### Study outcome used in granting a NOC/c

From January 1, 2005 documents on Health Canada's Summary Basis of Decision (SBD) website (https://hpr-rps.hres.ca/reg-content/summary-basis-decision-result.php?lang=en&term=) have explained why the agency authorized drugs for sale in Canada. One of those documents describes the pivotal studies that were submitted and the primary outcomes in those studies. On the basis of the descriptions, the outcomes were classified as either surrogate or clinical and the type of outcome was recorded.

All information was extracted by a single person and entered in an Excel spreadsheet.

### Data analysis

Counts were made of the total number of drugs that were approved with a NOC/c, how many did and did not fulfill their conditions and the median number of days from granting the NOC/c until fulfillment or until May 18, 2022 for drugs that had not fulfilled their conditions. The total number of studies listed in the QNs and their characteristics were counted. The characteristics of the confirmatory studies was analyzed at the level of the QNs and at the level of the individual studies. The type of outcome used to grant the NOC/c for all drugs and for oncology drugs in particular was compared to the expected type of outcome for studies mentioned in the QNs.

### Patients and ethics

All data was publicly available and gathered between May 18–21, 2022 except for the QNs and Letters of Undertaking that were previously obtained through Access to Information for a previous project. Ethics approval was not required.

## Results

There were NOC/c for 78 drugs; 39 conditions were fulfilled, 34 were unfulfilled, 4 drugs were discontinued by the company and the indication was withdrawn for 1 drug. In the case where the indication was withdrawn it was because confirmatory studies were negative (S1 File). The median number of days until conditions were fulfilled was 1200 (interquartile range (IQR) 829, 2201) and for drugs with unfulfilled conditions the median was 990 (IQR 333, 1866) days. QNs and/or Letters of Undertaking were missing for 2 drugs (in response to the ATI, in one case only an amendment to QN was sent and in the other a company announcement about the drug was sent) and all information was redacted from one QN. The 75 available QNs listed a total of 154 studies with a median of 2 studies (interquartile range 1, 3) per QN. Fifty-one drugs were antineoplastics, 7 were for the treatment of HIV and the remaining 20 were for a variety of other conditions. S1 File gives all the data that were used in this study.

### Methodology in studies in individual QNs

An absence of information meant that the study outcome was unknown in all the studies mentioned in 43 (57.3%) QNs, only surrogate outcomes were used in studies in 12 (16.0%) QNs, there was a mixture of surrogate and unknown outcomes in studies in 10 (13.3%) QNs, clinical and surrogate outcomes were used in studies in 7 (9.3%) QNs, clinical and unknown outcomes were used in studies in 2 (2.7%) QNs and clinical only outcomes were used in studies in 1 (1.3%) QN (Table 1).

**Table 1. Study methodology in individual Qualifying Notices (QN).**

| | Outcome (n = 75) | | | | | |
|---|---|---|---|---|---|---|
| | Unknown | All surrogate | Surrogate and unknown | Clinical and surrogate | Clinical and unknown | All clinical |
| Number of QNs | 43 (57.3%) | 12 (16.0%) | 10 (13.3%) | 7 (9.3%) | 2 (2.7%) | 1 (1.3%) |
| | Randomization (n = 75) | | | | | |
| | Unknown | All randomized | Randomized and unknown | Randomized, not randomized, unknown | All unrandomized | Randomized and not randomized |
| Number of QNs | 36 (48.0%) | 18 (24.0%) | 15 (20.0%) | 3 (4.0%) | 2 (2.7%) | 1 (1.3%) |
| | Blinding (n = 75) | | | | | |
| | Unknown | All unblinded | No blinding and unknown | All blinded | Blinded and unblinded | Blinded and unknown | Blinded, unblinded, unknown |
| Number of QNs | 42 (56.0%) | 10 (13.3%) | 10 (13.3%) | 7 (9.3%) | 3 (4.0%) | 2 (2.7%) | 1 (1.3%) |

Whether studies were randomized was unknown because of the absence of information in all the studies in 36 (48.0%) QNs, all studies were randomized in 18 (24.0%) QNs, there was a mixture of randomized and randomization unknown studies in 15 (20.0%) QNs, a mixture of randomized, not randomized and randomization unknown studies in 3 (4.0%) QNs, no randomized studies in 2 (2.7%) QNs and a mixture of randomized and not randomized studies in 1 (1.3%) QN (Table 1).

Blinding was unknown for any study in 42 (56.0%) QNs because of the absence of information, no studies were blinded in 10 (13.3%) QNs, there was a mixture of unblinded and blinding unknown in 10 (13.3%) QNs, all studies were blinded in 7 QNs, there was a mixture of blinded and unblinded studies in 3 (4.0%) QNs, a mixture of blinded and blinding unknown studies in 2 (2.7%) QNs and a mixture of blinded, unblinded and blinding unknown in 1 (1.3%) QN (Table 1). Only 21 (28.0%) QNs contained any information about how long studies should last and in only 12 (16.0%) of these 21 cases did all the studies in the QN have this information.

## Methodology in individual studies

From the total of 154 studies, the type of outcome to be used was unknown because of a lack of information in 99 (64.3%) studies, 11 (71.4%) confirmatory studies used clinical outcomes and 44 (28.6%) used surrogate outcomes. Randomization was unknown because of a lack of information in 96 (62.3%) studies, 51 (33.1%) were randomized and 7 (4.5%) were not. Blinding was unknown because of a lack of information in 97 (63.0%) studies, 36 (23.4%) were not blinded and 21 (13.6%) were (Table 2). The length of the study was given in 29 (18.8%) cases but in 2 cases the "trial timeline" was redacted from the information in the QN.

## Patient demographics in individual QNs

No QN had any information about the number of men and women in studies. Quantitative information about patient numbers was present in some studies in 15 (20.0%) QNs but only 6 (8.0%) QNs had this information in all studies. Patient ages were present in at least some studies in 11 (14.7%) QNs but only present in all studies in a QN in 7 (9.3) cases.

## Patient demographics in individual studies

The distribution of men and women was not given in 147 studies (the other 7 were for either breast, endometrial or ovarian cancer), the number of patients was given in 23 (14.9%) studies

**Table 2. Distribution of individual studies by outcome, randomization and blinding.**

| Aspect of methodology | | Number of studies (n = 154) |
|---|---|---|
| **Outcome** | **Clinical** | 11 (71.4%) |
| | **Surrogate** | 44 (28.6%) |
| | **Unknown** | 99 (64.3%) |
| **Randomization** | **Yes** | 51 (33.1%) |
| | **No** | 7 (4.5%) |
| | **Unknown** | 96 (62.3%) |
| **Blinding** | **Yes** | 21 (13.6%) |
| | **No** | 36 (23.4%) |
| | **Unknown** | 97 (63.0%) |

but in 5 cases vague terms such as "large number" or "sufficient number" were used. The age of patients was given in 24 (15.6%) studies (1 study said that there should be a subpopulation aged greater than 65), but in 15 (9.7%) cases relatively vague terms such as "children", "pediatric", "adolescent" and "adult" were used. No study gave information about race or ethnicity.

## Other information

The expected time to completion was available for 36 (23.4%) of the 154 of the confirmatory studies. Information to identify studies (some combination of identification number, full study title, acronym of study title) was present for 77 (50.0%) studies, absent for 23 (14.9%) and unclear for 26 (16.9%). In the other 28 (18.2%) cases, the study identification was recorded as being removed from the QN. In some QNs, even the name of the company marketing the drug was removed by Health Canada.

## Comparison between outcome used to approve drug and outcome of confirmatory studies

Sixty-four of the 75 drugs with QNs were approved after January 1, 2005 and therefore it was possible to use the SBD website to determine the outcome used in granting a NOC/c: 54 (84.4%) had surrogate outcomes, 4 (6.3%) had clinical outcomes, 5 (7.8%) were not found on the SBD website and one drug was approved too recently for there to be any documentation about the outcome used for approval. QNs were available for 58 drugs whose approval outcome was known so that the approval outcome and the outcome from confirmatory studies could be compared. The QNs did not contain the expected confirmatory trial outcome information for any of the 4 drugs approved with clinical outcomes. For the 54 drugs approved with surrogate outcomes, 8 (14.8%) confirmatory studies were expected to use clinical outcomes, 15 (27.8%) surrogate outcomes and no outcomes were mentioned for 31 (57.4%) (Table 3). Forty-five of the drugs with QNs and that were listed on the SBD website were for the treatment of cancer and all except one were approved on the basis of surrogate outcomes. Eight (18.2%) of the 44 oncology drugs approved with surrogate outcomes had confirmatory studies that used clinical outcomes, 14 (31.8%) used surrogate outcomes and the outcome could not be determined for 22 (50.0%). The sole oncology drug approved with a clinical outcome had a confirmatory study with a surrogate outcome (Table 3).

## Discussion

This cross-sectional study of 75 Qualifying Notices issued by Health Canada for drugs approved through its NOC/c pathway from July 1998 to May 18, 2022 found that the QNs

**Table 3. Outcome used to grant Notice of Compliance with conditions (NOC/c) and outcome to be used in confirmatory studies.**

| Drug indication | Outcome used to grant NOC/c (n) | | Outcome to be used in confirmatory study | | |
|---|---|---|---|---|---|
| | | | Clinical (n) | Surrogate (n) | Information not in Qualifying Notice (n) |
| All indications | Clinical (n) | 4 | 0 | 0 | 4 (100%) |
| | Surrogate (n) | 54 | 8 (14.8%) | 15 (27.8%) | 31 (57.4%) |
| Oncology | Surrogate (n) | 44 | 8 (18.2%) | 14 (31.8%) | 22 (50.0%) |
| | Clinical (n) | 1 | 0 | 0 | 1 (100%) |

contained little information about aspects of the methodology or patient demographics in confirmatory studies. The only partial exception to this lack of information was for study identification where the majority (77 out of 154) had some identifying material, but even here Health Canada had removed identifying information for 28 studies. The results of this study are broadly in-line with an examination of postmarket study descriptions of required studies for drugs approved by the US Food and Drug Administration between 2009 and 2012. In that case, of 110 required clinical trials, 38 (34.5%), 44 (40.0%), 62 (56.4%), 66 (60.0%), and 98 (89.1%) did not report enough information to establish use of randomization, comparator type, allocation, outcome, and number of patients to be enrolled, respectively [10].

The inconsistent inclusion and exclusion of information in the QNs is in-line with Health Canada's practice in the type of information included in the first phase of the SBD documents and also with its decisions about how to apply its criteria for which drugs qualify for a NOC/c. An analysis of SBDs up to April 2012 found that clinical trial information was presented in a haphazard manner. At least one-third of the potential information about patient trial characteristics and the benefits and risks of tested treatments was missing [11]. McPhail and colleagues concluded that eligibility criteria for oncology drugs to qualify for a NOC/c are insufficiently defined, resulting in their inconsistent application by Health Canada [12].

During the median 1200 days (3.29 years) that it takes for the studies to be conducted and evaluated by Health Canada and for the conditions to be fulfilled, prescribers and patients have little idea about how much additional information these confirmatory studies will add to what is already known about the drug. Once the studies have been completed, there is only enough identifying information about half of them mentioned in the QNs to allow interested parties to search for published versions. The paucity of information that Health Canada provides about confirmatory studies in the QNs means that it would be difficult to identify published versions of those studies and to compare them to Health Canada's requirements to see if manufacturers had adhered to those requirements. Other work that examined published confirmatory studies specified in QNs found that while most were randomized (80.6%) over half (55.6%) used surrogate outcomes and only slightly more than a third (34.5%) were blinded. Patients tended to be relatively young even for drugs approved for the treatment of cancer and men outnumbered women [13].

Especially concerning is that only 8 of the 44 oncology drugs approved using surrogate outcomes were being tested in confirmatory studies with clinical outcomes, whereas 14 were being tested in studies with further surrogate outcomes. (In 22 cases the type of outcome was not specified.) In a systematic review of studies that attempted to validate surrogate outcomes in oncology products, 65 specific surrogate survival pairs were identified. Of these, 52% were classified as low strength, 25% as medium strength and only 23% correlated highly with overall survival [14]. Gyawali and colleagues [15] found that out of 93 oncology drugs approved by the Food and Drug Administration, 19 confirmatory studies reported improvement in the same surrogate used in the preapproval trial, and 20 (21%) reported improvement in a different

surrogate. Based on their analysis, they conclude that reassessment of the requirements for confirmatory trials may be necessary to obtain more clinically meaningful information.

## Limitations

A protocol for this study was not registered in advance. All the data was gathered by a single person and therefore there were no methods for reducing biases such as inter-rater agreement. The QNs contained other types of information, for example about the frequency of reporting of safety issues, and some QNs contained detailed descriptions of how drugs should be studied in confirmatory studies, but that information was not examined. Outcomes used in approving 16 drugs prior to 2005 were not available and QNs were missing or information was redacted for 3 drugs.

## Conclusion

At present, Health Canada provides minimal information about the requirements for confirmatory studies in the QNs. That means that in most cases clinicians have little knowledge about Health Canada's requirements for confirmatory studies, whether those studies will provide the necessary information to resolve questions about the drug's benefits and whether manufacturers are adhering to those requirements. As a first step, Health Canada needs to develop a validated list of what information regarding confirmatory studies should be contained in a QN and ensure that this information is included in a consistent manner. Health Canada should also reconsider whether further studies with surrogate outcomes provide sufficient information about efficacy when drugs, especially those for oncology, are conditionally approved using surrogate outcomes.

## Supporting information

**S1 File. Complete data collected.**
(XLSX)

**S1 Checklist. STROBE statement—Checklist of items that should be included in reports of observational studies.**
(DOCX)

## Author Contributions

**Conceptualization:** Joel Lexchin.

**Data curation:** Joel Lexchin.

**Formal analysis:** Joel Lexchin.

**Methodology:** Joel Lexchin.

**Writing – original draft:** Joel Lexchin.

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
