## [Decision Letter · Decision Letter 0]

21 Jul 2022

PONE-D-22-15922Information about confirmatory studies required for new drugs conditionally approved by Health Canada: a cross-sectional studyPLOS ONE

Dear Dr. Lexchin,

Thank you for submitting your manuscript to PLOS ONE. After careful consideration, we feel that it has merit but does not fully meet PLOS ONE’s publication criteria as it currently stands. Therefore, we invite you to submit a revised version of the manuscript that addresses the points raised during the review process.

First of all, **I would like to thank the two reviewers for their important feedback**. I'm following their opinion and suggest major revisions. as both agree on this recommandation. Please also note that one reviewer was very critical about the study. I agree that more details are needed, especially concerning the methods and the results. Of course, you are expected to take all the comments from the 2 reviewers into consideration in your revised manuscript but please have a very strong focus on the methods and the results, in order to make sure that all important aspects for reproducibility purpose are present. The use of one or more reporting guideline may help (see on the EQUATOR NETWORK). In case more material needs to be share, it can be attached as a supplement or posted on the Open Science Framework.  Please also note that without an appropriate consideration of the major edits that are required, I cannot guarantee publication. Please also make it explicit if a protocol was registered for this study (and where) and in case, no, please add a few words about that in the limitation section.  Please also write a few words about the main limitation in the abstract to avoid any spin. 

We look forward to receiving your revised manuscript.

Kind regards,

Florian Naudet, M.D., M.P.H., Ph.D.

Academic Editor

PLOS ONE

Journal Requirements:

In 2019-2021, Joel Lexchin received payments for writing a brief on the role of promotion in generating prescriptions for Goodmans LLP and from the Canadian Institutes of Health Research for presenting at a workshop on conflict-of-interest in clinical practice guidelines. He is a member of the Foundation Board of Health Action International and the Board of Canadian Doctors for Medicare. He receives royalties from University of Toronto Press and James Lorimer & Co. Ltd. for books he has written.  

Reviewers' comments:

Reviewer's Responses to Questions

**Comments to the Author**

1. Is the manuscript technically sound, and do the data support the conclusions?

Reviewer #1: Partly

Reviewer #2: No

2. Has the statistical analysis been performed appropriately and rigorously? 

Reviewer #1: Yes

Reviewer #2: N/A

3. Have the authors made all data underlying the findings in their manuscript fully available?

Reviewer #1: Yes

Reviewer #2: No

4. Is the manuscript presented in an intelligible fashion and written in standard English?

Reviewer #1: Yes

Reviewer #2: Yes

5. Review Comments to the Author

Reviewer #1: This study is a cross-sectional study assessing the quality of the reporting of data regarding the design of confirmatory studies that were requested after a conditonal approval granted by Health Canada (under the Notice of Compliance with condition (NOC/c) policy). Information (patient demographics, design, outcomes, expected date of completion, record of information for allowing the retrival of published data of the completed confirmatory studies) at the drug and study level are supposed to be provided under the Qualifying Notices policy. Data were retrieved for drugs for which a NOC/c was granted since the begining of the program until May 2022 (with request for filling a Qualifying Notice (QN) starting in 2003 and the primary outcome for approving new drugs recorded since 2005). The study also compares the type of primary outcome (i.e., clinical, surrogate) between the study that led to NOC/c and the expected outcome in the confirmatory studies. The main result is the fact that information are frequently lacking regarding most aspects of confirmatory studies (patient demographics, study designs, outcomes) and confirmatory studies are frequently eexpected to be conducted on surrogate outcomes. In its current stage, the reported information hampers the possibility for clinicians and patients to understand if confirmatory studies will alleviate or not uncertainties regarding treatment efficacy. The conclusion is the need for a standardized reporting of these aspects by Health Canada.

Overall, the methodology of the study seems sound, but I had troubles understanding what is the specific definition of the sample of drugs that were analyzed and I had troubles to understand if the info that were analyzed were info about what the confirmatory studies were expected to be before completion, or info about how confirmatory studies were indeed conducted. I'm confident the confusion I've experienced is just a matter of clarification withhin the text and not due to design flaws, so I'm confident it will be easily resolved. However, currently, I was not able to fully understand what truly the findings of the study are about. I will try to pinpoint below why I've experienced this confusion.

The study analyzes 78 drugs with NOC/c that were granteds since the begining of the program. It is specified it is ddivided into 39 drugs for which the condition of the program were "fullfilled", 34 were "unfulfilled" and 5 drugs were discontinued from the market before fullfilling their conditions. But what does "fullfilled" and "unfullfilled" really mean?

At first, I was under the impression "fullfilled" means the confirmatory study was completed AND the drug was granted a full NOC. Thus, "unfullfilled" means the confirmatory study was completed AND the drug failed to obtain a full NOC. But, tha data analysis section indicates "the median number of days from granting the NOC/c until fullfillment or until May 18, 2022 (date of database lock) for drugs that had not fullfilled their conditions was estimated". Therefore, does it means "fullfilled" only describes drugs for which confirmatory studies were completed whether the results were positive or not, and "unfullfilled" only describes drugs for which the confirmatory study is still under conduct? This aspect needs to be clarified for understanding what is the definition of the "population" analyzed in the study. Of note: I have looked reference nuimber 6 of the paper which indicates a sample of 89 new drugs and new indications with NOC/c untile May 2017. I understand the present study does not analyze new indications, but still why only 78 drugs if this study covers a period untile May 2022?

Moreover, I was confused about the nature of the info that are recorded under under the Qualifying Notices policy. Does those info are only about what will be the confirmatory studies as provided by the health technology manufacturer when granted a NOC/c (therefore prospective info about what will be done) or is it retrospective info on how the confirmatory studies were indeed conducted, or is it a mix of both?

Indeed, the study tackles data such as expected date of completion (therefore info about what the confirmatory study will be conducted), but also compares outcomes of the initial study leading to NOC/c and outcomes expected in confirmatory studies (the word expected seems to imply the info for confirmatory studies are info about what will be expected and not what has been done, but I was not sure which one it really was). But, the study also tackles info about patient demographics and sample size. Here, I was under the impression the study is about info on how the confirmatory studies were really conducted and collected after completion. In the end, I think the paper needs to clarify what's what in order to be fully able to understand the results.

I will now discuss more specific details.

I/INTRODUCTION

The date of the begining of the NOC/c program could be clearly specified.

Page 5. After the first paragraph, I think the eligibility criteria a drug needs to fullfill for the NOC/c program could be specified in accordance with the Health Canada website in addition to the goal of the program.

Page 5. It is specified that before February 2003 QN did not exist and info were provided into a letter of understanding. Therefore, does it makes sense to analyze drugs since the begining of the NOC/c program instead of the begining of the request for a QN? Was it expected the same info were requested under the letter of understanding policy?

Page 6. Please, specify why QNs for new indications were excluded from the study.

II/ METHODS

What was the criteria for deciding an outcome was a clinical or surrogate one?

On a related note, the method should clearly states data were only retrieved and analyze by one person, therefore there were no methods for reducing measurement biases such as inter-rater agreement.

III/ RESULTS

In general, does "undetermined" characteristics (such as study outcome, randomization) corresponds to missing data, or data that were not reported with sufficient clarity for understanding the info, or a mix of both? I think the paper could benefit from a little more detail about this aspect.

Reviewer #2: The paper authored by Joel Lexchin throws a glance on regulatory decisions and requirements following conditional or accelerated approval of drugs in Canada. In a cursory approach, 78 conditional approval decisions by Health Canada were retrieved from the regulator’s website and other sources, which are not detailed. Names of the specific drugs, year of conditional approval, or the manufacturer are not provided.

This article does not contribute new insights to the scientific debate on how shifting clinically important efficacy and safety assessments from before to after authorization affects our ability to obtain an unbiased assessment of the risk-benefit ratio and safety of newly developed drugs (1-4). Meanwhile, there are many reports that post-authorisation studies often fail to deliver—lots of studies are never started, many take years longer than planned, and some fail to confirm pre-authorisation results. Evidence on relevant outcomes often remains inconclusive for several years, (5-7) and post-authorisation safety events are seen more frequently for drugs with expedited approval.(8) The sketchy findings presented by Lexchin suggest that Health Canada seems to accept that the manufacturers are not adhering to post-authorisation study requirements, and drugs are only rarely withdrawn, as was described before.(2)

The paper has major limitations. Methods, results and conclusions are not well founded. It is unclear how drugs approved under the NOC/c policy were identified and how exactly the information on QNs was retrieved and compiled in which time period. The tables are not clear and uninformative to the reader, totals are missing. In table 1, the calculation of total study numbers is N=70 for “Outcome” and “Randomization” but 69 for “Blinding”. A descriptive table of availability of main characteristics for each of the 78 QNs (e.g. number of studies, study identification, duration of conditional approval procedures, drug name and class, number, age and sex of participants, date of conditional approval, and outcome measures in NOCs and QN) is not provided. The result section is vague and does not attempt to look for potential associations, e.g. between missing information and drug class or duration of conditional approval.

The conclusion to call for a regulatory template for “confirmatory studies” is weak. Does the author think that manufacturers would better adhere to regulatory templates than to QNs and why should they? Would regulators be more restrictive by using the templates instead the QNs and why should they? Rather, the integrity of accelerated/conditional approval should be discussed. Is there really a justification for curtailing or totally omitting Phase III trials ?

Take Covid-19 mRNA vaccines as an example: After a median of only 2 months follow-up, the pivotal Phase III studies were unblinded and placebo-group participants were offered vaccination, thereby introducing bias which may severely impair the comparability of vaccinated and non-vaccinated participants in the remaining observation time of the study. In light of the many unknowns of Covid-19 mRNA vaccine efficacy and safety, stopping the randomized trials at the time of conditional/ emergency approval violates basic principles of good medical practice. (9) A very recent independent analysis of adverse events summary tables submitted to the FDA for emergency use approval shows that the risk of severe adverse events of special interest after mRNA vaccination exceeds the benefit risk reduction of hospitalization due to Covid-19. The authors call for a “systematic review and meta-analysis using individual participant to address questions of harm-benefit in various demographic subgroups. Full transparency of the COVID-19 vaccine clinical trial data is needed to properly evaluate these questions. Unfortunately, well over a year after widespread use of COVID-19 vaccines, participant level data remain inaccessible.” (10) Therefore, my conclusion is to omit conditional approval wherever possible. Access to Individual participant data (IPD) should be granted for independent scrutiny for all drug approval pathways.

1) Dawoud D, Naci H, Ciani O, Bujkiewicz S. Raising the bar for using surrogate endpoints in drug regulation and health technology assessment. BMJ2021;374:n2191. doi:10.1136/bmj.n2191

2) Naci H, Smalley KR, Kesselheim AS. Characteristics of preapproval and postapproval studies for drugs granted accelerated approval by the US Food and Drug Administration. JAMA2017;318:626-36. doi:10.1001/jama.2017.9415

3) Moore TJ, Furberg CD. Development times, clinical testing, postmarket follow-up, and safety risks for the new drugs approved by the US food and drug administration: the class of 2008. JAMA Intern Med2014;174:90-5. doi:10.1001/jamainternmed.2013.11813

4) Darrow JJ, Avorn J, Kesselheim AS. FDA approval and regulation of pharmaceuticals, 1983-2018. JAMA2020;323:164-76. doi:10.1001/jama.2019.20288

5) Davis C, Naci H, Gurpinar E, Poplavska E, Pinto A, Aggarwal A. Availability of evidence of benefits on overall survival and quality of life of cancer drugs approved by European Medicines Agency: retrospective cohort study of drug approvals 2009-13. BMJ2017;359:j4530. doi:10.1136/bmj.j4530

6) Beaver JA, Howie LJ, Pelosof L, et al. A 25-year experience of US Food and Drug Administration accelerated approval of malignant hematology and oncology drugs and biologics: a review. JAMA Oncol2018;4:849-56. doi:10.1001/jamaoncol.2017.5618

7) Gyawali B, Hey SP, Kesselheim AS. Assessment of the clinical benefit of cancer drugs receiving accelerated approval. JAMA Intern Med2019;179:906-13. doi:10.1001/jamainternmed.2019.0462

8) Downing NS,Shah ND, Aminawung JA, et al. Postmarket safety events among novel therapeutics approved by the US Food and Drug Administration Between 2001 and 2010. JAMA2017;317:1854-63. doi:10.1001/jama.2017.5150

9) Prugger, C., Spelsberg A., Keil U. et al. Evaluating covid-19 vaccine efficacy and safety in the post-authorisation phase. BMJ 2021;375:e067570

10) Fraiman, J. Erviti J, Jones M. et al. Serious adverse events of special interest following mRNA vaccination in randomized trials. https://ssrn.com/abstract=4125239

6. PLOS authors have the option to publish the peer review history of their article (what does this mean?). If published, this will include your full peer review and any attached files.

Reviewer #1: No

Reviewer #2: **Yes: **Angela Spelsberg

---

## [Author Response · Author response to Decision Letter 0]

8 Aug 2022

Dear Dr. Naudet:

Thank you and the reviewers for your comments. I have responded to each of the comments below and look forward to hearing from you.

Sincerely

Joel Lexchin MD

Comments from the academic editor:

I agree that more details are needed, especially concerning the methods and the results. Of course, you are expected to take all the comments from the 2 reviewers into consideration in your revised manuscript but please have a very strong focus on the methods and the results, in order to make sure that all important aspects for reproducibility purpose are present. 

The Methods and Results sections have been revised taking into consideration the comments from the reviewers.

The use of one or more reporting guideline may help (see on the EQUATOR NETWORK). In case more material needs to be share, it can be attached as a supplement or posted on the Open Science Framework. 

The Strobe checklist is included in the submission of the revised manuscript.

Please also make it explicit if a protocol was registered for this study (and where) and in case, no, please add a few words about that in the limitation section. 

A statement has been added to the limitations that there was no protocol registered in advance.

Please also write a few words about the main limitation in the abstract to avoid any spin. 

The Discussion section in the Abstract now states “All of the data were gathered by a single individual possibly introducing unintended biases.”

Reviewer #1: 

Overall, the methodology of the study seems sound, but I had troubles understanding what is the specific definition of the sample of drugs that were analyzed and I had troubles to understand if the info that were analyzed were info about what the confirmatory studies were expected to be before completion, or info about how confirmatory studies were indeed conducted. I'm confident the confusion I've experienced is just a matter of clarification within the text and not due to design flaws, so I'm confident it will be easily resolved. 

I trust that my responses below will clarify any confusion that the reviewer had.

However, currently, I was not able to fully understand what truly the findings of the study are about. I will try to pinpoint below why I've experienced this confusion. The study analyzes 78 drugs with NOC/c that were granted since the beginning of the program. It is specified it is divided into 39 drugs for which the condition of the program were "fulfilled", 34 were "unfulfilled" and 5 drugs were discontinued from the market before fulfilling their conditions. But what does "fulfilled" and "unfulfilled" really mean? At first, I was under the impression "fulfilled" means the confirmatory study was completed AND the drug was granted a full NOC. Thus, "unfulfilled" means the confirmatory study was completed AND the drug failed to obtain a full NOC. But, the data analysis section indicates "the median number of days from granting the NOC/c until fulfillment or until May 18, 2022 (date of database lock) for drugs that had not fulfilled their conditions was estimated". Therefore, does it means "fulfilled" only describes drugs for which confirmatory studies were completed whether the results were positive or not, and "unfulfilled" only describes drugs for which the confirmatory study is still under conduct? This aspect needs to be clarified for understanding what is the definition of the "population" analyzed in the study.

In the Introduction unfulfilled is now defined as either the confirmatory studies not having been completed or being completed but not satisfactory to Health Canada and fulfilled is defined as completed and satisfactory to Health Canada.

Of note: I have looked reference number 6 of the paper which indicates a sample of 89 new drugs and new indications with NOC/c until May 2017. I understand the present study does not analyze new indications, but still why only 78 drugs if this study covers a period until May 2022?

Out of the 89 NOC/c analyzed in reference 6, 52 were for new drugs and 37 were for new indications of existing drugs. Therefore, between May 2017 and May 2022 there were an additional 26 NOC/c issued for new drugs. The number of NOC/c issued annually is variable and depends on manufacturers’ submissions.

Moreover, I was confused about the nature of the info that are recorded under the Qualifying Notices policy. Does those info are only about what will be the confirmatory studies as provided by the health technology manufacturer when granted a NOC/c (therefore prospective info about what will be done) or is it retrospective info on how the confirmatory studies were indeed conducted, or is it a mix of both? Indeed, the study tackles data such as expected date of completion (therefore info about what the confirmatory study will be conducted), but also compares outcomes of the initial study leading to NOC/c and outcomes expected in confirmatory studies (the word expected seems to imply the info for confirmatory studies are info about what will be expected and not what has been done, but I was not sure which one it really was).

But, the study also tackles info about patient demographics and sample size. Here, I was under the impression the study is about info on how the confirmatory studies were really conducted and collected after completion. In the end, I think the paper needs to clarify what's what in order to be fully able to understand the results.

The Introduction points out that the current study uses the perspective of clinicians in examining the information about the confirmatory studies in the QNs. Specifically, will these studies provide clinicians with knowledge that will help resolve uncertainties about the benefits of drugs with a NOC/c: a) what are the demographics of the patients who will be enrolled; b) what methodology will the studies employ (e.g., randomization and blinding); and c) if the drugs were given a NOC/c based on surrogate outcomes will the confirmatory studies use clinical outcomes. In addition, are two other pieces of information that are potentially important to clinicians in the QNs: when will the studies be completed so that they can take necessary precautions in prescribing until that point is there enough identifying information about the studies that will enable clinicians to search for and read the studies after they have been completed.

I will now discuss more specific details.

INTRODUCTION

The date of the beginning of the NOC/c program could be clearly specified.

The Introduction now states that the NOC/c pathway started in 1998.

Page 5. After the first paragraph, I think the eligibility criteria a drug needs to fulfill for the NOC/c program could be specified in accordance with the Health Canada website in addition to the goal of the program.

The Introduction now says that confirmatory studies are evaluated similar to any studies done before a drug is marketed and provides a reference to the relevant Health Canada document.

Page 5. It is specified that before February 2003 QN did not exist and info were provided into a letter of understanding. Therefore, does it makes sense to analyze drugs since the begining of the NOC/c program instead of the begining of the request for a QN? Was it expected the same info were requested under the letter of understanding policy?

This study evaluated all NOC/c issued since the start of the pathway. The Introduction now points out that the Letter of Undertaking needs to outline commitments for confirmatory studies that will be done in accordance with established scientific standards and the timelines for these studies that will verify the clinical benefit of the drug. Therefore, it is expected that the Letter of Undertaking should contain the same information as the information in the QN.

Page 6. Please, specify why QNs for new indications were excluded from the study.

QNs for new indications for existing drugs were not analyzed for a number of reasons: a) Health Canada does not provide the date when the application for the new indication was filed; b) there is no publicly available source about whether surrogate or clinical outcomes were used in granting a NOC/c for the new indication; and c) when a new indication is granted clinicians will have had some experience prescribing the drug and therefore the information in the confirmatory studies may be less important for them.

METHODS

What was the criteria for deciding an outcome was a clinical or surrogate one?

From January 1, 2005 documents on Health Canada’s Summary Basis of Decision website (https://hpr-rps.hres.ca/reg-content/summary-basis-decision-result.php?lang=en&term=) have explained why the agency authorized drugs for sale in Canada. One of those documents describes the pivotal studies that were submitted and the primary outcomes in those studies. On the basis of the descriptions, the outcomes were classified as either surrogate or clinical and the type of outcome was recorded.

On a related note, the method should clearly states data were only retrieved and analyze by one person, therefore there were no methods for reducing measurement biases such as inter-rater agreement.

The Methods section already states that all of the information was gathered by a single person and the Limitations section repeats that information. 

RESULTS

In general, does "undetermined" characteristics (such as study outcome, randomization) corresponds to missing data, or data that were not reported with sufficient clarity for understanding the info, or a mix of both? I think the paper could benefit from a little more detail about this aspect.

The term “undetermined” has been replaced by “unknown” and it is made clear that a result is unknown because of a lack of information.

Reviewer #2: 

The paper authored by Joel Lexchin throws a glance on regulatory decisions and requirements following conditional or accelerated approval of drugs in Canada. In a cursory approach, 78 conditional approval decisions by Health Canada were retrieved from the regulator’s website and other sources, which are not detailed. 

I am not sure why the reviewer considers this to be a cursory approach but if s/he could provide more details I would be happy to try and respond.

Names of the specific drugs, year of conditional approval, or the manufacturer are not provided.

This information, along with all the other information extracted is now provided in a supplementary file.

This article does not contribute new insights to the scientific debate on how shifting clinically important efficacy and safety assessments from before to after authorization affects our ability to obtain an unbiased assessment of the risk-benefit ratio and safety of newly developed drugs (1-4). 

The type of analysis that the reviewer is proposing is certainly valid and one approach would be through a narrative review or preferably a systematic review. One step in understanding how regulator required post-market studies affect our understanding of safety and efficacy compared to what was learned from pre-market studies is knowing what type of information the regulators have required from the post-market studies, e.g., are the studies asking the correct questions, studying the right populations, using strong methodology. The current study is attempting to answer some of those questions.

Meanwhile, there are many reports that post-authorisation studies often fail to deliver—lots of studies are never started, many take years longer than planned, and some fail to confirm pre-authorisation results. Evidence on relevant outcomes often remains inconclusive for several years, (5-7) and post-authorisation safety events are seen more frequently for drugs with expedited approval.(8) The sketchy findings presented by Lexchin suggest that Health Canada seems to accept that the manufacturers are not adhering to post-authorisation study requirements, and drugs are only rarely withdrawn, as was described before.(2)

I agree with the reviewer that drugs with an expedited approval have more post-authorization safety events than drugs with a standard review. In fact, one of my previous studies demonstrates that (Lexchin J. Post-market safety warnings for drugs approved in Canada under the Notice of Compliance with conditions policy. British Journal of Clinical Pharmacology 2015;79:847-859.) I also agree with the reviewer that relatively few drugs are withdrawn because their confirmatory have failed to show efficacy. Again, that is a topic that I have previously explored (Lexchin J. A comparison of the Food and Drug Administration’s and Health Canada’s regulatory decisions about failed confirmatory trials for oncology drugs: an observational study. Journal of Pharmaceutical Policy and Practice 2021;14:93. https://doi.org/10.1186/s40545-021-00375-y.) Contrary to what the reviewer implies, my study does not suggest that Health Canada accepts that manufacturers are not adhering to post-authorization study requirements. That may well be the case, but that would require identifying and analyzing the post-market studies and comparing the information in them to what Health Canada had initially required. However, that task would be difficult for two reasons: as my study shows, knowing what Health Canada has required and identifying published versions of post-authorization studies would both be very difficult to accomplish because of the paucity of information that Health Canada provides in the QNs. I now identify those consequences of the lack of information in the Discussion.

The paper has major limitations. Methods, results and conclusions are not well founded. It is unclear how drugs approved under the NOC/c policy were identified and how exactly the information on QNs was retrieved and compiled in which time period. 

The Methods section clearly states that drugs with a NOC/c were identified from one of three sources: the NOC/c website, a previous study of mine (Lexchin J. Notice of Compliance with conditions: a policy in limbo. Healthcare Policy. 2007;2:114-22) and a study by Law (Law M. The characteristics and fulfillment of conditional prescription drug approvals in Canada. Health Policy. 2014;116:154-61.) QNs were downloaded from the NOC/c website and if they were not available there then an Access to Information request was used. If drugs were approved with a NOC/c before QNs were being used then an Access to Information request was made for the relevant Letter of Undertaking. The Methods section now states that all of the information was gathered between May 18-21 except for the QNs and Letters of Undertaking that were previously acquired through Access to Information.

The tables are not clear and uninformative to the reader, totals are missing. In table 1, the calculation of total study numbers is N=70 for “Outcome” and “Randomization” but 69 for “Blinding”.

I thank the reviewer for pointing out this error which has now been corrected.

A descriptive table of availability of main characteristics for each of the 78 QNs (e.g. number of studies, study identification, duration of conditional approval procedures, drug name and class, number, age and sex of participants, date of conditional approval, and outcome measures in NOCs and QN) is not provided.

Suppementary File 1 now provides all the information that was used in the study.

The result section is vague and does not attempt to look for potential associations, e.g. between missing information and drug class or duration of conditional approval.

The reviewer seems to be asking for a regression analysis which would presumably use date of NOC/c, drug class and time until fulfillment of conditions as independent variables. However, there are multiple potential dependent variables. If the reviewer could specify which dependent variables s/he thinks are important I could explore the value of doing regression analysis.

The conclusion to call for a regulatory template for “confirmatory studies” is weak. Does the author think that manufacturers would better adhere to regulatory templates than to QNs and why should they? Would regulators be more restrictive by using the templates instead the QNs and why should they? Rather, the integrity of accelerated/conditional approval should be discussed. Is there really a justification for curtailing or totally omitting Phase III trials? Take Covid-19 mRNA vaccines as an example: After a median of only 2 months follow-up, the pivotal Phase III studies were unblinded and placebo-group participants were offered vaccination, thereby introducing bias which may severely impair the comparability of vaccinated and non-vaccinated participants in the remaining observation time of the study. In light of the many unknowns of Covid-19 mRNA vaccine efficacy and safety, stopping the randomized trials at the time of conditional/ emergency approval violates basic principles of good medical practice. (9) A very recent independent analysis of adverse events summary tables submitted to the FDA for emergency use approval shows that the risk of severe adverse events of special interest after mRNA vaccination exceeds the benefit risk reduction of hospitalization due to Covid-19. The authors call for a “systematic review and meta-analysis using individual participant to address questions of harm-benefit in various demographic subgroups. Full transparency of the COVID-19 vaccine clinical trial data is needed to properly evaluate these questions. Unfortunately, well over a year after widespread use of COVID-19 vaccines, participant level data remain inaccessible.” (10) Therefore, my conclusion is to omit conditional approval wherever possible. Access to Individual participant data (IPD) should be granted for independent scrutiny for all drug approval pathways.

As per my answer to one of the reviewer’s previous comments, I agree that it would be important to have a much stronger understanding of what Health Canada requires in confirmatory studies, whether those requirements will provide the necessary information to resolve questions about benefits and whether manufacturers are adhering to those requirements. Therefore, as a first step in this process I think that a call for Health Canada to develop a template of what information should be included in the QNs about confirmatory studies is a reasonable requirement. Once that step is taken, then we can start to answer the questions posed above about confirmatory studies in particular and the NOC/c pathway in general. 

The reviewer uses the problems with the Phase III study of the Pfizer COVID vaccine to argue against omitting pre-market Phase III studies which is what the NOC/c process does. Again, I agree with the reviewer that omitting Phase III studies often leads to an overestimating of the benefits of drugs and an underestimation of harms as the FDA report “22 case studies where phase 2 and phase 3 trials had divergent results”. Arguing for the elimination of NOC/c and similar pathways is valid but that was not the purpose of my study.

---

## [Decision Letter · Decision Letter 1]

28 Sep 2022

PONE-D-22-15922R1Information about confirmatory studies required for new drugs conditionally approved by Health Canada: a cross-sectional studyPLOS ONE

Dear Dr. Lexchin,

Thank you for submitting your manuscript to PLOS ONE. After careful consideration, we feel that it has merit but does not fully meet PLOS ONE’s publication criteria as it currently stands. Therefore, we invite you to submit a revised version of the manuscript that addresses the points raised during the review process.

Sorry for the delay in answering, but -as I have answered to your information request to PLOS One last month- one of the reviewer required more time to perform his review for personal reasons that I totally understand. In addition, I needed the feedback from this reviewer to arbitrate as the other reviewer was quite unhappy with your revisions and asked to reject the paper. The second reviewer who has just answered, as promised, asked for minor revisions. I am also satisfied with the changes made in response to my own comments. However I wanted to discuss the case with PLOS One staff before making my final decision, facing those 2 divergent reviews. It took a few additional extra days. After this process, and considering the comments of the 2 reviewers, I ask you a new round of major revisions. Indeed, I do think that the comments by the 2 reviewers can be fixed by a new round of revisions and that the points raised by the critical reviewer can be addressed. I would be therefore very happy to consider a new version of this manuscript. I will likely invite a new reviewer for the next round. I agree that it is a long journey for a paper but, I also think that it will make it even better. Importantly, I would like to thank both reviewers for their comments.

We look forward to receiving your revised manuscript.

Kind regards,

Florian Naudet, M.D., M.P.H., Ph.D.

Academic Editor

PLOS ONE

Reviewers' comments:

Reviewer's Responses to Questions

**Comments to the Author**

1. If the authors have adequately addressed your comments raised in a previous round of review and you feel that this manuscript is now acceptable for publication, you may indicate that here to bypass the “Comments to the Author” section, enter your conflict of interest statement in the “Confidential to Editor” section, and submit your "Accept" recommendation.

Reviewer #1: (No Response)

Reviewer #2: (No Response)

2. Is the manuscript technically sound, and do the data support the conclusions?

Reviewer #1: Yes

Reviewer #2: No

3. Has the statistical analysis been performed appropriately and rigorously? 

Reviewer #1: Yes

Reviewer #2: N/A

4. Have the authors made all data underlying the findings in their manuscript fully available?

Reviewer #1: (No Response)

Reviewer #2: Yes

5. Is the manuscript presented in an intelligible fashion and written in standard English?

Reviewer #1: Yes

Reviewer #2: Yes

6. Review Comments to the Author

Reviewer #1: Overall, the revised version of the draft has alleviated the issues that I have raised in my previous review.

Minor remarks:

Introduction: I suggest to reword the sentence "Most approvals are based on promising surrogate outcomes" by "Most approvales are based on surrogate outcomes assumed to be validated for surrogacy". Indeed, it is discussed later that the validity of numerous surrogate outcomes can be doubtfull. However, the current version of the sentence seems to imply all surrogate outcomes are promising.

Results section: There are still occurence within the manuscript and tables of the term "undetermined" instead of "unknown".

Discussion page 17: There may be a typo in the sentence "whereas 14 were being tested in studies with further surrogate outcomes". Should it be 44 instead of 14?

Conclusion: The new begining of the conclusion could be modified a little to be more accurate regarding the findings. Indeed, the findings are striking regarding the lack of information in the QNs, but maybe the conclusion is not as strong as "virtually no one... has any knowledge...." because sometimes there are information.

Reviewer #2: Thank you for revising the manuscript and addressing critical points, particularly providing the supplementary table with key information on the 78 drugs conditionally approved between 1998 and 2022 . From this table follows that:

1. The -use of conditional approval pathways is increasing

There were only 34 NOCs issued during the first 17 years between 1998 and 2014, i.e. roughly 2 per year. In contrast, between 2015 and 2021, 43 drugs were conditionally approved, about 6 drugs per year

2. Conditionally approved drugs are increasingly focusing on lucrative cancer treatment

From 1998 through 2014 there were 6 NOC/c on drugs for HIV treatment, 19 for cancer, and 9 for the treatment of mixed conditions. During 2015-2021, all 43 except one were oncology drugs.

3. Drug withdrawals in conditionally approved pathways decrease

Four of 34 conditionally approved drugs between 1998 and 2014 were withdrawn, among them the prominent example natrecor ®, an expensive „innovative“ drug for acute congestive heart failure which was shown to be a killer of patients by independent scientific evaluation of primary trial reports to the FDA and from the manufacturer (Sackner-Bernstein JD et al. JAMA 2005;293:1900-1905).

After 2015, only one drug out of 43 was withdrawn by the manufacturer.

From the perspective of clinicians, it would be more informative to describe which 8 drugs had missing QNs, which drugs or drug class had no information on outcome (38), randomization (33) and blinding (39), for which drugs or drug class the number of patients and their ages (23) and surrogate outcomes were available(52) and for which drugs or drug class outcomes could not be determined (30).

According to the author, the aim of the paper is “to resolve uncertainties about the benefits of drugs with a NOC/c”. I do not see how the proposed templates can basically improve the sobering reality of conditional/expedited approval pathways. The rationale for ommitting phase III pivotal trials i.e. to provide innovative treatments faster to patients with life-threatening illnesses must be questioned. Innovation does not automatically imply more good than harm. The conditional approval pathway puts patients, doctors, and society at high risks of no or marginal benefit and severe undiscovered adverse events at extremely high costs. In my opinion, scientists have to speak up against this bad clinical practice jointly committed by regulators and manufacturers.

7. PLOS authors have the option to publish the peer review history of their article (what does this mean?). If published, this will include your full peer review and any attached files.

Reviewer #1: No

Reviewer #2: **Yes: **Angela Spelsberg

---

## [Author Response · Author response to Decision Letter 1]

2 Oct 2022

I want to thank the academic editor for his careful and thorough management of my submission and the reviewers for their insightful comments. Below I indicate how I have responded to the individual comments.

Reviewer #1: 

Overall, the revised version of the draft has alleviated the issues that I have raised in my previous review. 

I am glad that my revisions have met the reviewer’s standards.

Minor remarks: 

Introduction: I suggest to reword the sentence "Most approvals are based on promising surrogate outcomes" by "Most approvals are based on surrogate outcomes assumed to be validated for surrogacy". Indeed, it is discussed later that the validity of numerous surrogate outcomes can be doubtful. However, the current version of the sentence seems to imply all surrogate outcomes are promising. 

I have added a direct quotation from the Health Canada guidance document on its Notice of Compliance with conditions policy and the sentence now reads “Most approvals are based on promising surrogate outcomes that “are reasonably likely, based on available evidence, to predict an effect of a drug on recognized clinical outcomes such as morbidity and mortality.”

Results section: There are still occurence within the manuscript and tables of the term "undetermined" instead of "unknown". 

All instances of the word “undetermined” have been deleted.

Discussion page 17: There may be a typo in the sentence "whereas 14 were being tested in studies with further surrogate outcomes". Should it be 44 instead of 14? 

14 is the correct number but to avoid confusion I have added that the type of outcome in 23 cases was not specified.

Conclusion: The new beginning of the conclusion could be modified a little to be more accurate regarding the findings. Indeed, the findings are striking regarding the lack of information in the QNs, but maybe the conclusion is not as strong as "virtually no one... has any knowledge...." because sometimes there are information.

The start of the sentence now reads “That means that in most cases clinicians have little knowledge…”

Reviewer #2: Thank you for revising the manuscript and addressing critical points, particularly providing the supplementary table with key information on the 78 drugs conditionally approved between 1998 and 2022 . From this table follows that: 

1. The -use of conditional approval pathways is increasing. There were only 34 NOCs issued during the first 17 years between 1998 and 2014, i.e. roughly 2 per year. In contrast, between 2015 and 2021, 43 drugs were conditionally approved, about 6 drugs per year.

2. Conditionally approved drugs are increasingly focusing on lucrative cancer treatment. From 1998 through 2014 there were 6 NOC/c on drugs for HIV treatment, 19 for cancer, and 9 for the treatment of mixed conditions. During 2015-2021, all 43 except one were oncology drugs. 

The reviewer has pointed out some very interesting trends in the use of the NOC/c pathway

and I have previously published about these changes (up to the end of 2017), please see:

Lexchin J. Health Canada’s use of its Notice of Compliance with Conditions drug approval

policy: a retrospective cohort analysis. International Journal of Health Services 2019:49:294

305.

3. Drug withdrawals in conditionally approved pathways decrease. Four of 34 conditionally approved drugs between 1998 and 2014 were withdrawn, among them the prominent example natrecor ®, an expensive „innovative“ drug for acute congestive heart failure which was shown to be a killer of patients by independent scientific evaluation of primary trial reports to the FDA and from the manufacturer (Sackner-Bernstein JD et al. JAMA 2005;293:1900-1905). After 2015, only one drug out of 43 was withdrawn by the manufacturer. 

I did not have the resources to do an extensive search for the reasons why drugs were withdrawn but I did search the NOC/c website to see what was recorded for these drugs. There was no information about 3 of them and for one the reason recorded was that the confirmatory studies did not demonstrate efficacy. This information is now mentioned in the text and I have added a column to Supplementary File 1 with this information.

From the perspective of clinicians, it would be more informative to describe which 8 drugs had missing QNs, which drugs or drug class had no information on outcome (38), randomization (33) and blinding (39), for which drugs or drug class the number of patients and their ages (23) and surrogate outcomes were available (52) and for which drugs or drug class outcomes could not be determined (30). 

After further searching I was able to find 5 of the 8 missing QNs and have now included the information in them in the text and tables. (The inclusion of this additional information meant that many of the figures had to be changed.) In addition, I have now clarified in the text that all information was redacted from one QN and in the remaining two QNs the wrong documents were sent in response to the Access to Information request.

Interested readers can find the information about which drugs contained missing information in the QNs from Supplementary File 1. It is possible that there was some pattern to the exclusion of information but it is much more likely that the exclusion was the result of the inconsistency in how Health Canada constructed these documents. This inconsistency can also be found in the information in Summary Basis of Decision documents and in decisions about which drugs qualify for a NOC/c. These inconsistencies are now noted at the end of the first paragraph of the Discussion. 

According to the author, the aim of the paper is “to resolve uncertainties about the benefits of drugs with a NOC/c”. I do not see how the proposed templates can basically improve the sobering reality of conditional/expedited approval pathways. The rationale for omitting phase III pivotal trials i.e., to provide innovative treatments faster to patients with life-threatening illnesses must be questioned. Innovation does not automatically imply more good than harm. The conditional approval pathway puts patients, doctors, and society at high risks of no or marginal benefit and severe undiscovered adverse events at extremely high costs. In my opinion, scientists have to speak up against this bad clinical practice jointly committed by regulators and manufacturers.

With respect to the reviewer, the aim of the study was not “to resolve uncertainties about the benefits of drugs with a NOC/c”. That would require examining the information in completed confirmatory studies and then determining if that information was sufficient to more definitively determine benefits and harms in the populations that were using the drugs. The aim of my study as stated in the Introduction was to examine the quantity and quality of the information in the Qualifying Notices. 

As I said in my previous response to the reviewer, I agree that omitting Phase III studies often leads to an overestimating of the benefits of drugs and an underestimation of harms as the FDA report “22 case studies where phase 2 and phase 3 trials had divergent results” shows. However, I think that it is necessary to have a strong evidentiary basis to argue for the retention of Phase III studies in the drug approval process. Part of that evidentiary basis is having a much stronger understanding of the limitations of the NOC/c pathway. One aspect of that understanding is an examination of the requirements for confirmatory studies and whether those requirements will provide the necessary information to resolve questions about the benefits of drugs approved and whether manufacturers are adhering to those requirements. A first step in this process is a call for Health Canada to develop a template of what information should be included in the QNs. Once that step is taken, then we can start to answer questions posed about the value of confirmatory studies in particular and the NOC/c pathway in general.

---

## [Decision Letter · Decision Letter 2]

10 Oct 2022

PONE-D-22-15922R2Information about confirmatory studies required for new drugs conditionally approved by Health Canada: a cross-sectional studyPLOS ONE

Dear Dr. Lexchin,

Thank you for submitting your manuscript to PLOS ONE. After careful consideration, we feel that it has merit but does not fully meet PLOS ONE’s publication criteria as it currently stands. Therefore, we invite you to submit a revised version of the manuscript that addresses the points raised during the review process. As I have said previously, I have invited a new reviewer to assess the manuscript because of the discrepancies between the 2 previous reviewers. I really would like to tank the reviewer for being so fast in reviewing the manuscript. **He made a few comments that can be easily addressed in a round of revisions. I look forward to reading your manuscript.**

We look forward to receiving your revised manuscript.

Kind regards,

Florian Naudet, M.D., M.P.H., Ph.D.

Academic Editor

PLOS ONE

Journal Requirements:

Reviewers' comments:

Reviewer's Responses to Questions

**Comments to the Author**

1. If the authors have adequately addressed your comments raised in a previous round of review and you feel that this manuscript is now acceptable for publication, you may indicate that here to bypass the “Comments to the Author” section, enter your conflict of interest statement in the “Confidential to Editor” section, and submit your "Accept" recommendation.

Reviewer #3: (No Response)

2. Is the manuscript technically sound, and do the data support the conclusions?

Reviewer #3: Yes

3. Has the statistical analysis been performed appropriately and rigorously? 

Reviewer #3: N/A

4. Have the authors made all data underlying the findings in their manuscript fully available?

Reviewer #3: Yes

5. Is the manuscript presented in an intelligible fashion and written in standard English?

Reviewer #3: Yes

6. Review Comments to the Author

Reviewer #3: Thank you for the opportunity to review this manuscript on the information about confirmatory studies required for new drugs conditionally approved by Health Canada. Overall, this manuscript is methodologically sound, easy to read and understand, and suitable for publication. However, I have a few comments that will further increase clarity:

Abstract Methods:

“A list of drugs approved under the NOC/c policy and their QNs were sourced” .. please report from where?

Abstract Results

- Provide time-frame for 78 drugs approved using NOC/c (dates)

- Provide median number of studies per drug with IQR

- Should say proportion of men and women?

- Where is information about other aspects of methodology that are considered primary endpoints?

Abstract Discussion

“Confirmatory studies with surrogate outcomes were often used to validate efficacy in drugs initially approved using surrogate outcomes” – often may not be the right word, considering it is <30%.

Introduction:

- Missing reference for “provide patients suffering from serious….”

- The author mentions that prior to 2/2003 the QN did not exist? The abstract and results section of the manuscript can be improved by more transparent reporting of dates

- Some of the objectives in the introduction are not discussed in the abstract – when will the studies be completed and is there enough information about the studies that will enable them to search for and read the studies

- The final paragraph of the introduction seems like a methods section paragraph

Results:

- The reporting of characteristics at the QN level is rather confusing. I think the following reporting system could improve clarity:

At least one study randomized vs. no studies randomized

At least one study blinded vs. no studies blinded

At least one clinical outcome vs. no clinical outcomes

- Table 1 is a little difficult to follow. Please see suggestion above.

Methodology in individual studies

- Please report %’s (e.g. 99 (X%)) for the numerators

- Table 2, please report %’s

Patient demographics

- What about race/ethnicity or age?

Other information

- Please report %’s

Discussion

- In the opening of the discussion, please recap study design, time, and sample. E.g.. In this cross-section study of XXXXX…..between XXX and XXXXX,

- The opening of the discussion highlights an outcome that was not included in the abstract and therefore does not appear to be a major finding – the information to identify studies.

- What are SBD documents?

- The author could consider referencing https://www.bmj.com/content/361/bmj.k2031 , which focuses on the lack of information including in descriptions of postmarketing requirements for FDA-approved drugs. (disclosure: I am an author on this manuscript, so please do not feel any pressure to reference this paper. I just thought it had some similar and interesting findings).

7. PLOS authors have the option to publish the peer review history of their article (what does this mean?). If published, this will include your full peer review and any attached files.

Reviewer #3: No

---

## [Author Response · Author response to Decision Letter 2]

10 Oct 2022

I am grateful to the reviewer for evaluating the manuscript so rapidly. Below I explain how I have dealt with the comments. 

Reviewer #3: Thank you for the opportunity to review this manuscript on the information about confirmatory studies required for new drugs conditionally approved by Health Canada. Overall, this manuscript is methodologically sound, easy to read and understand, and suitable for publication. However, I have a few comments that will further increase clarity.

I thank the reviewer for her/his comments and am happy to hear that s/he felt that the manuscript was suitable for publication.

Abstract Methods 

“A list of drugs approved under the NOC/c policy and their QNs were sourced” .. please report from where? 

The following has been added to the end of the sentence “from two previous publications as well as Health Canada’s NOC/c website”.

Abstract Results

• Provide time-frame for 78 drugs approved using NOC/c (dates) 

The following was added “from the time the first drug was approved under the program in July 1998 until May 18, 2022”.

• Provide median number of studies per drug with IQR

The following was added “median of 2 studies per QN, interquartile range 1,3”.

• Should say proportion of men and women?

The abstract already states that “No study gave the distribution of men and women”.

• Where is information about other aspects of methodology that are considered primary endpoints?

The following was added to the end of the Results section “Specifically for oncology drugs, 44 were approved with surrogate outcomes and one with a clinical outcome. Eight (18.2%) of the 44 oncology drugs approved with surrogate outcomes had confirmatory studies that used clinical outcomes, 14 (31.8%) used surrogate outcomes and the outcome could not be determined for 22 (50.0%). The sole oncology drug approved with a clinical outcome had a confirmatory study with a surrogate outcome”.

Abstract Discussion 

• “Confirmatory studies with surrogate outcomes were often used to validate efficacy in drugs initially approved using surrogate outcomes” – often may not be the right word, considering it is <30%.

“Often” was replaced by “almost one-third of the time”.

Introduction

• Missing reference for “provide patients suffering from serious….”

Reference 2 applies to this statement and has been added.

• The author mentions that prior to 2/2003 the QN did not exist? The abstract and results section of the manuscript can be improved by more transparent reporting of dates

The start of the first sentence in the Methods now reads “The generic name of all new drugs approved with a NOC/c along with the dates of granting and fulfillment of the NOC/c from the time that the first drug was approved under the program in July 1998 until May 18, 2022…”. 

As indicated above, the same information is now included in the Results section of the Abstract. 

I am not sure what other dates the reviewer would like me to include but I would be happy to add them if the reviewer can be more specific.

• Some of the objectives in the introduction are not discussed in the abstract – when will the studies be completed and is there enough information about the studies that will enable them to search for and read the studies

The following sentence was added to the Results section of the Abstract “The expected time of completion of the studies was available for 36 out of 154 and information to identify studies was present for 77, absent for 23 and unclear for 26.”

• The final paragraph of the introduction seems like a methods section paragraph.

The paragraph that the reviewer refers to has been shifted to the end of the “Information Extraction” subsection of the Methods.

Results 

• The reporting of characteristics at the QN level is rather confusing. I think the following reporting system could improve clarity:

o At least one study randomized vs. no studies randomized

o At least one study blinded vs. no studies blinded

o At least one clinical outcome vs. no clinical outcomes

The type of reporting suggested by the reviewer seems to assume that there is a binary result, e.g., yes or no, but there are multiple results for whether there was randomization, blinding and type of outcome. If the reviewer has a suggestion for a way to report multiple results, I would be happy to reconsider rewording.

• Table 1 is a little difficult to follow. Please see suggestion above.

The suggestion above seems to be for the text not for the table, but percentages have been added to the numbers in the table and in the text.

Methodology in individual studies

• Please report %’s (e.g., 99 (X%)) for the numerators

Percentages have now been added.

• Table 2, please report %’s

Percentages have been added to the table and the text.

Patient demographics

• What about race/ethnicity or age?

There was no information reported about race/ethnicity for any study and that information has been added in the Results. The Results already report information about age: “The age of patients was given in 24 (15.6%) studies (1 study said that there should be a subpopulation aged greater than 65), but in 15 (9.7%) cases relatively vague terms such as “children”, “pediatric”, “adolescent” and “adult” were used.”

• Other information

The limitations subsection already says “The QNs contained other types of information, for example about the frequency of reporting of safety issues, and some QNs contained detailed descriptions of how drugs should be studied in confirmatory studies, but that information was not examined.”

• Please report %’s

Percentages have been added to all the numbers.

Discussion

• In the opening of the discussion, please recap study design, time, and sample. E.g.. In this cross-section study of XXXXX…..between XXX and XXXXX 

The opening part of the first sentence of the Discussion now reads “This cross-sectional study of 75 Qualifying Notices issued by Health Canada for drugs approved through its NOC/c pathway from July 1998 to May 18, 2022 found that the QNs…”

• The opening of the discussion highlights an outcome that was not included in the abstract and therefore does not appear to be a major finding – the information to identify studies.

The following information has been added to the Results in the Abstract “information to identify studies was present for 77, absent for 23 and unclear for 26”.

• What are SBD documents?

SBD documents are already described in the subsection of the Methods “Study outcome used in granting a NOC/c”.

• The author could consider referencing https://www.bmj.com/content/361/bmj.k2031 , which focuses on the lack of information including in descriptions of postmarketing requirements for FDA-approved drugs. (disclosure: I am an author on this manuscript, so please do not feel any pressure to reference this paper. I just thought it had some similar and interesting findings).

I am grateful to the reviewer for drawing this paper to my attention. I have cited it in the Discussion.

---

## [Editor Report · Decision Letter 3]

12 Oct 2022

Information about confirmatory studies required for new drugs conditionally approved by Health Canada: a cross-sectional study

PONE-D-22-15922R3

Dear Dr. Lexchin,

We’re pleased to inform you that your manuscript has been judged scientifically suitable for publication and will be formally accepted for publication once it meets all outstanding technical requirements. **Thank you for taking the time to address all the reviewer's comments. Please note a typo in table 1 that you will have to correct at the proof stages (a parenthesis ")" is missing after 20%). **

Kind regards,

Florian Naudet, M.D., M.P.H., Ph.D.

Academic Editor

PLOS ONE
---

## [Editor Report · Acceptance letter]

13 Oct 2022

PONE-D-22-15922R3 

Information about confirmatory studies required for new drugs conditionally approved by Health Canada: a cross-sectional study 

Dear Dr. Lexchin:

I'm pleased to inform you that your manuscript has been deemed suitable for publication in PLOS ONE. Congratulations! Your manuscript is now with our production department. 

Kind regards, 

on behalf of

Pr. Florian Naudet 

Academic Editor

PLOS ONE